

# Effects of two-week e-learning on eHealth literacy: a randomized controlled trial of Japanese Internet users

Toshiharu Mitsuhashi

Center for Innovative Clinical Medicine, Okayama University Hospital, Okayama, Japan

## ABSTRACT

**Background**. The Internet is widely used as a source of information by people searching for medical or healthcare information. However, information found on the Internet has several drawbacks, and the ability to consume accurate health information on the Internet (eHealth literacy) is increasingly important. This study's goal was to clarify the extent to which eHealth literacy is improved after e-learning in a randomized controlled trial.

**Methods**. Data were collected on 301 Japanese adults through an online survey. Participants were assigned to the intervention (e-learning about eHealth literacy) group or the control group in a 1:1 ratio. The intervention group included 148 participants, and 153 participants were in the control group. The participants provided information at baseline on demographic characteristics, self-rated health, and frequency of Internet searching. The eHealth Literacy Scale (eHEALS), which was the main measure of eHealth literacy, and data on secondary outcomes (the Healthy Eating Literacy Scale and skill for evaluating retrieved search results) were obtained at baseline and at follow-up. The score difference was calculated by subtracting the score at baseline from the score at follow-up. Linear regression analysis and multinomial regression analysis were performed using the differences in score as the dependent variables and the intervention as the explanatory variable. Intention-to-treat analysis was employed.

**Results**. The results from participants who responded to all of the questions both times were analyzed (134 in the intervention group and 148 in the control group). eHEALS increased 1.57 points due to the intervention effect ($\Delta$ score change = 1.57; 95% CI [0.09–3.05]; $p = 0.037$). Skills for evaluating retrieved search results improved more in the intervention group than in the control group (relative risk ratio = 2.47; 95% Confidence Interval: 1.33, 4.59; $p = 0.004$). There were no large differences at baseline between the intervention and control groups in the eHEALS, Healthy Eating Literacy scale, or skill for evaluating retrieved search results. However, at follow-up, the intervention group had improved more than the control group on both the eHEALS and skill for evaluating retrieved search results.

**Discussion**. eHealth literacy improved after the e-learning, as evidenced by the change to the eHEALS scores and increased skill for evaluating retrieved search results. There was no significant effect of e-learning, which did not include content on healthy eating, on the Healthy Eating Literacy Scale scores. This indicates that scores did not increase much due to effects other than e-learning, as is sometimes seen with the Hawthorne effect. Although it was statistically significant, the effect size was small. Therefore, future research is necessary to verify the clinical implications. In sum, this study suggests that e-learning is an effective way to improve eHealth literacy.

Corresponding author
Toshiharu Mitsuhashi,
mitsuh-t@cc.okayama-u.ac.jp

# INTRODUCTION

The public widely uses the Internet as a source of medical and healthcare information. However, information found on the Internet has several drawbacks (*Zhang, Sun & Xie, 2015*). First, available or retrieved information might be incomplete (*De Groot et al., 2017*; *Takegami et al., 2017*). Second, the information as written might not be clear (*Daraz et al., 2018*; *De Groot et al., 2017*). Third, even scientifically reliable information is not highly ranked in search engine results unless Search Engine Optimization is performed (*Modave et al., 2014*). Fourth, some problems with software tools that help users to organize and make sense of health information exist (*Hernández et al., 2017*). Fifth, the assessment tools of health information have important limitations (*Beaunoyer et al., 2017*). Therefore, scientifically reliable websites might not be retrieved, suggesting that information found on the Internet is not sufficient to obtain scientific reliability and reliance on it might actually be harmful to health (*Bizzi, Ghezzi & Paudyal, 2017*; *Kothari & Moolani, 2015*).

Because of the unreliability of online information on health, it is important that people have the ability to critically appraise the health information that they obtain from the Internet. The skill involved with that ability is referred to as "health literacy," and is generally defined as "the ability to correctly examine and utilize health-related information" (*Ad Hoc Committee on Health Literacy for the Council on Scientific Affairs, American Medical Association, 1999*; *Nutbeam, 1998*; *Sørensen et al., 2012*). However, as evidenced by several surveys, the public's level of health literacy is not high. According to a German survey, 54.3% of respondents were found to have limited health literacy (*Schaeffer, Berens & Vogt, 2017*). In a survey from England, 52% of respondents did not have an adequate score (*Protheroe et al., 2017*). According to a 2015 Japanese survey, about 85.4% of the respondents had health literacy problems (*Nakayama et al., 2015*). Thus, research indicates that health literacy is low on a global scale.

However, Internet use rises every year, and it is increasingly important for the public to be able to obtain accurate information from the Internet for healthcare decision-making. *Norman & Skinner (2006a)* dubbed this ability "eHealth literacy" and defined it as "the ability to seek, find, understand, and appraise health information from electronic sources and apply the knowledge gained to addressing or solving a health problem" (p. 1). Since then, investigation of eHealth literacy has been limited, but survey results have found that people with low eHealth literacy might be likely to be exposed to incorrect or incomplete health information, which has been related to adverse health outcomes (*De Boer, Versteegen & Van Wijhe, 2007*). Therefore, education to improve eHealth literacy is important to public health.

Some previous studies have found that eHealth literacy improved after educational interventions (*Robinson & Graham, 2010*; *Xie, 2011a*; *Xie, 2011b*). However, these studies had study design problems that interfered with the ability to demonstrate the effects

of educational interventions. For example, a control group was not included and/or participants were not randomly assigned. Moreover, the influence of e-learning on eHealth literacy has not been studied. Therefore, this study aimed to clarify the extent to which eHealth literacy is influenced by e-learning in a randomized controlled trial in Japan.

## MATERIALS & METHODS

### Ethical considerations for studies on human subjects

This study was approved by the Okayama University Graduate School of Medicine, Dentistry and Pharmaceutical Sciences and Okayama University Hospital, Ethics Committee (approval number K1707-025). The study was not registered because it does not meet the International Committee of Medical Journal Editors' criteria of a clinical experiment, and the study's results do not directly relate to patient outcomes. The purpose and method of research and experiment were appropriately described to potential participants on the recruitment webpage. After this description, informed consent was obtained from participants. They were free to refuse to participate for any reason.

### Trial design

This study was a parallel, Internet-based, randomized controlled trial (RCT) of health literacy educational intervention by e-learning. First, a baseline questionnaire survey was administered online between September 29, 2017, and October 3, 2017. Then, the participants were 1:1 assigned to the intervention and control groups. The group receiving the treatment was exposed to e-learning for eHealth literacy during the 14 days from October 10, 2017, to October 23, 2017. A follow-up online questionnaire survey was administered from October 23, 2017 through October 30, 2017. This paper reports on the study using a modified Consolidated Standards of Reporting Trials (CONSORT) guideline checklist (http://www.consort-statement.org).

#### *Randomization*

After the baseline survey was completed, the participants were assigned to the intervention group or the control group using stratified block randomization with a block size of four in a 1:1 ratio. The participants were sorted into four strata by gender and age because both characteristics relate to eHealth literacy (*Halwas, Griebel & Huebner, 2017*; *Mitsutake et al., 2012*). The participants were assigned to their groups by an automated system using Stata do-file mechanism, and, therefore, the investigator was not aware of, and did not personally participate in the group assignments. However, both groups could not be blinded.

### Participants

This study's 300 participants were recruited from the population of about 1.2 million registered members of Macromill, Inc., which is a Japanese online survey company (https://monitor.macromill.com/). The participants were recruited from the member pool using four strata of 75 participants each: males aged 20 to 39 years, males aged 40 to 59 years, females aged 20 to 39 years, and females aged 40 to 59 years. The inclusion criteria were: (1) agreement to participate, (2) interest in e-learning, and (3) interest in health literacy. There were no exclusion criteria.

Recruitment was conducted from September 14 through 19, 2017. When the number of participants reached 300, recruitment was terminated. Because two participants simultaneously applied, the total sample size was 301. The sampling process is shown in Fig. 1. Data on gender, age, residence, household income, and frequency of Internet search activity were collected in the baseline questionnaire. The participants were randomly assigned to the intervention ($n = 148$) or the control ($n = 153$) group after they completed the baseline questionnaire. Ultimately, 282 pieces of participant data were analyzed (134 in the intervention group and 148 in the control group) because 19 participants dropped out before the follow-up.

All participants who answered every question were given 100 tokens (JPY 100, USD .94), and all of the participants who answered every question and completed the e-learning content were given 1,000 tokens (JPY 1,000, USD 9.36).

### Sample size calculation

It was assumed that the primary outcome, eHealth Literacy Scale (eHEALS) scores of the intervention group, would improve by 2.0 points compared to the control group. In a previous study (*Mitsutake et al., 2011*), the standard deviation of eHEALS was 6.45. Because the participant data were considered similar to each other with respect to the inclusion criteria, the eHEALS scores were assumed to vary less and, therefore, the standard deviation was assumed to be smaller than previously found. It was expected to be about 6.0, and it was determined that $\alpha = 0.05$ and $\beta = 0.20$. Under these conditions, the required sample size was calculated as 143 per group. Considering sample attrition, the sample size was set at 150 per group.

## Trial intervention

The intervention comprised e-learning content created by the researcher. Text material of e-learning content has been prepared as a supplementary file. The content was presented to the participants in simple Japanese to facilitate comprehension. The content included text and images on the following topics: (1) reliability of information on the Internet, (2) scientific research methods, and (3) cautions regarding health information posted on social networking websites. The e-learning comprised 5,000 Japanese characters per topic. The entire e-learning content could easily be completed over a two-week period with about 10 min of dedicated application to learning the content per day. To confirm the participants' knowledge gained from the e-learning activity, four optional quizzes were included in the learning content.

## Outcomes (dependent variables)

All of the learning outcomes were measured using the participants' online responses to the baseline and follow-up questionnaires.

### Primary outcome (eHEALS)

The eHEALS is an eight-item self-report questionnaire that assesses knowledge, comfort, and perceived skill at finding, evaluating, and applying electronic health information to health problems (*Norman & Skinner, 2006b*). The response options on the items ranged

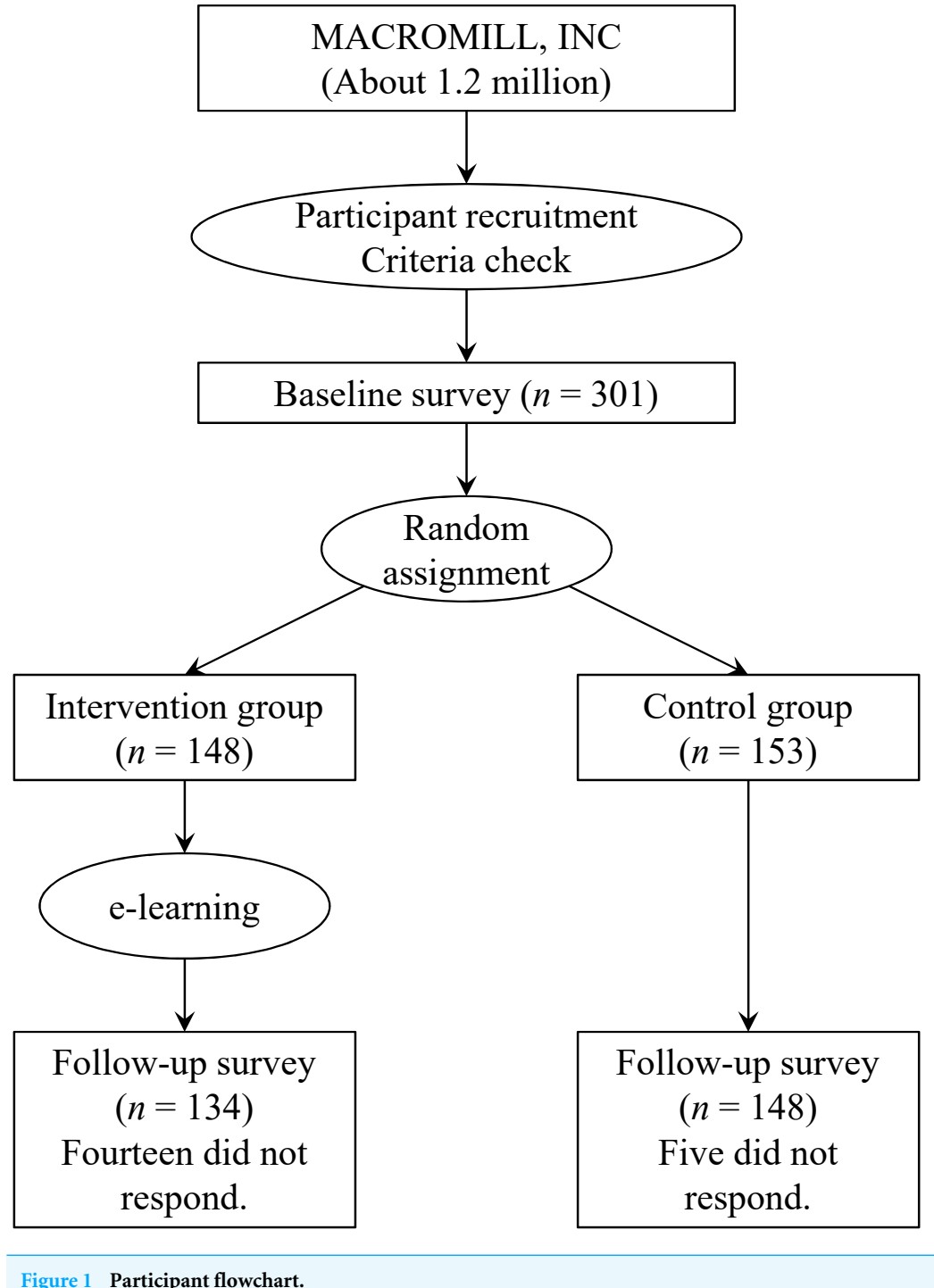

**Figure 1  Participant flowchart.**

from 1 = *not at all* to 5 = *strongly agree*. The responses on the items were summed, and these composite scores ranged from 8 to 40. The Japanese version of eHEALS was developed by *Mitsutake et al. (2011)*. In the sample, Cronbach's alpha was 0.916 at baseline and 0.913 at follow up.

### Secondary outcomes

This study assessed two secondary outcomes of e-learning: (1) the Healthy Eating Literacy scale (HEL), and (2) the skill for evaluating retrieved search results (evaluation skill).

The HEL is a five-item scale that measures interactive and critical literacy about healthy diet. The HEL was employed to assess change to health literacy other than change to eHealth literacy. The response options on the HEL's items ranged from 1 = *not at all* to 5 = *strongly agree*. Each subject was assigned a single score ranging from 1 to 5, which was the average of his or her responses on the five items. The HEL was developed by *Kanae et al. (2012)*. Cronbach's alpha was 0.867 at baseline and 0.794 at follow up.

The HEL scale was used to examine the Hawthorne effect (*Franke & Kaul, 1978*). Since the intervention group was observed more in detail, such as with the tracking of the number of e-learning logins and the overall login time, than was the control group, the score might have risen due to the Hawthorne effect (*McCarney et al., 2007*). Since the intervention group did not learn about healthy eating through the e-learning content, the HEL score should not rise simply because of the e-learning. If that score did rise, it was considered to be evidence of the Hawthorne effect.

The evaluation skill in this study was defined as the skill needed to evaluate the reliability of webpages from retrieved search results with a limited amount of information. The participants' evaluation skill was assessed using a question adapted from previous research (*Van Deursen & Van Dijk, 2009*; *Van Deursen & Van Dijk, 2010*; *Van Deursen & Van Dijk, 2011*). In a previous study from 2011, the health literacy performance test was conducted using a laptop computer in a university office to measure the four types of skills (operational, formal, information, and strategic internet skills). However, since web questionnaires were conducted in this research, it was difficult to measure operational, formal, and strategic internet skills. Therefore, in this study, information skills were used to measure evaluation skills.

For assessment of evaluation skills, the participants were shown a results page with five retrieved websites and asked to indicate which of the five websites should be viewed first. The search results page, which was created for this study, listed two commercial websites, two personal healthcare websites, and one governmental laboratory website. Search result summaries and URL type (co.jp, com, ne.jp, and go.jp) were presented for the participants to use in determining their choices. For the two commercial webpages, the URL types were co.jp and com; from the title and the summary, it could be judged that the webpages were created by the seller. For the two non-expert healthcare webpages, the URL types were co.jp and ne.jp, and from the title and the summary, it could be judged that the webpages were created by non-experts. The URL of the one governmental laboratory webpage was go.jp, and it was explicitly stated that on this website, experts create articles for accurate information dissemination in the results summary. The participants who selected the governmental laboratory were identified as having mastered the evaluation skill. The participants with the skill were assigned one point, and those without the skill were assigned zero points. Change between the baseline and the follow-up survey was computed by subtracting the baseline score from the follow-up scores. Calculation results

were $+1$, $0$, $-1$, which were defined as better, no change, and worse, respectively. This measure has not been validated.

## Statistical analysis

Participants who were in in the intervention group but did not learn the e-learning content were analyzed as an intervention group (Intention-to-treat analysis). Statistical analysis was performed using Stata (Stata Corporation, version 15, College Station, TX, USA).

### Descriptive statistics

Means and standard deviations were used to describe the normally distributed continuous variables, and medians and interquartile ranges were used to describe the non-normally distributed continuous variables. Categorical variables were described using proportional distributions.

### Inferential statistics

To estimate the influence of e-learning on eHEALS and HEL, differences between the scores before and after (after scores minus before scores) the intervention were calculated. Linear regression analyses were performed using the difference scores as the dependent variables and the intervention as the explanatory variable, yielding unstandardized regression coefficients and their 95% confidence intervals (CIs). Next, Cohen's d and its 95% CIs were calculated as the effect size.

To estimate the influence of the intervention on evaluation skill, multinomial logistic regression analysis was performed to regress the evaluation skill change on intervention, yielding relative risk ratios (RRR) and their 95% CIs using no change as the reference outcome (*Hamilton, 1993*). This model was selected since the dependent variable has more than two categories. The point estimate of RRR is calculated using the following equation.

$$RRR_{\text{outcome}=j} = \frac{P(\text{outcome}=j|\text{intervention})}{P(\text{outcome}=\text{no change}|\text{intervention})} \Bigg/ \frac{P(\text{outcome}=j|\text{control})}{P(\text{outcome}=\text{no change}|\text{control})}.$$

For the significance test of the unstandardized regression coefficient, the Wald statistic and its 95% CIs were calculated.

### Ancillary analysis

Missing data on the dependent variables due to non-response at follow-up were handled through multiple imputation by predictive means matching (*Morris, White & Royston, 2014*). The inferential analyses were performed on the complemented dataset ($n = 301$) as well as on the original dataset ($n = 282$).

Supplementarily, participants who were in in the intervention group but did not learn the e-learning content were excluded from the analysis (per-protocol analysis).

**Table 1  Baseline characteristics of the sample**

| Variable | Entire sample (*n* = 301) | Intervention group (*n* = 148) | Control group (*n* = 153) |
|---|---|---|---|
| Female (*n* (%)) | 150 (49.8) | 74 (50.0) | 76 (49.7) |
| Age in years (mean (SD)) | 40.2 (10.1) | 40.2 (9.9) | 40.2 (10.2) |
| Educational attainment of university or more (*n* (%)) | 178 (59.1) | 81 (54.7) | 97 (63.4) |
| Parental status (*n* (%)) | 133 (44.2) | 67 (45.3) | 66 (43.1) |
| Household income/year in JPY millions (*n* (%)) | | | |
|     Less than 2 | 19 (6.3) | 10 (6.8) | 9 (5.9) |
|     2 or more and less than 4 | 56 (18.6) | 30 (20.3) | 26 (17.0) |
|     4 or more and less than 6 | 78 (25.9) | 35 (23.6) | 43 (28.1) |
|     6 or more and less than 8 | 44 (14.6) | 25 (16.9) | 19 (12.4) |
|     8 or more and less than 10 | 22 (7.3) | 12 (8.1) | 10 (6.5) |
|     10 or more and less than 12 | 18 (6.0) | 7 (4.7) | 11 (7.2) |
|     12 or more and less than 15 | 3 (1.0) | 1 (0.7) | 2 (1.3) |
|     15 or more and less than 20 | 7 (2.3) | 4 (2.7) | 3 (2.0) |
|     20 or more | 3 (1.0) | 1 (0.7) | 2 (1.3) |
|     Unknown | 16 (5.3) | 9 (6.1) | 7 (4.6) |
|     Missing | 35 (11.6) | 14 (9.5) | 21 (13.7) |
| Marital status (*n* (%)) | | | |
|     Married | 161 (53.5) | 82 (55.4) | 79 (51.6) |
|     Never married | 120 (39.9) | 55 (37.2) | 65 (42.5) |
|     Divorced/widowed | 20 (6.6) | 11 (7.4) | 9 (5.9) |
| Employment status (*n* (%)) | | | |
|     Full-time | 163 (54.2) | 73 (49.3) | 90 (58.8) |
|     Part-time | 46 (15.3) | 29 (19.6) | 17 (11.1) |
|     Self-employed | 24 (8.0) | 15 (10.1) | 9 (5.9) |
|     Other | 4 (1.3) | 2 (1.4) | 2 (1.3) |
|     None | 64 (21.3) | 29 (19.6) | 35 (22.9) |
| Self-rated health (*n* (%)) | 250 (83.1) | 118 (79.7) | 132 (86.3) |
| Internet search engine use <once/day (*n* (%)) | 43 (14.3) | 19 (12.8) | 24 (15.7) |

## RESULTS

### Baseline characteristics

Table 1 shows the participants' characteristics at baseline. The differences between the intervention and control groups were small on most of the items. The proportion with university or more education was 54.7% in the intervention group and 63.4% in the control group. Self-rated health was 79.7% in the intervention group and 86.3% in the control group. Self-rated health is a single-item summary measure of the perception of one's health. It is one suitable method for measuring adult health status (*Boardman, 2006*).

Ten participants (6.8%) out of the intervention group did not complete the material. On average, they completed 63.2% of the e-learning content. Twenty-seven participants (18.2%) did not even start the material.

**Table 2** Means, standard deviations (SD), change scores (follow-up minus baseline), and intervention effects (Δ change)[a] compared to control group.

| Dependent variable | Value | Intervention group (*n* = 134) | Control group (*n* = 148) | Intervention v. Control Δ change[a] and Cohen's d (95% Confidence Interval) *p*-value |
|---|---|---|---|---|
| The eHealth Literacy Scale, eHEALS (mean (SD)) | Baseline | 24.5 (6.61) | 25.9 (6.18) | 1.57 (0.09, 3.05) |
| | Follow-up | 26.8 (5.84) | 26.6 (5.63) | 0.250 (0.01, 0.48) |
| | Score change | 2.31 (7.27) | 0.74 (5.25) | *p* = 0.037 |
| The Healthy Eating Literacy Scale, HEL (mean (SD)) | Baseline | 3.44 (0.71) | 3.52 (0.70) | −0.08 (−0.22, 0.07) |
| | Follow-up | 3.50 (0.63) | 3.65 (0.54) | −0.12 (−0.38, 0.11) |
| | Score change | 0.06 (0.65) | 0.14 (0.59) | *p* = 0.300 |

**Notes.**
[a]Score change of intervention group minus score change of control group.

**Table 3** Results on evaluation skill at baseline and follow-up, intervention effect, and comparison of intervention group to control group.

| Variable | Value | Intervention group (*n* = 134) | Control group (*n* = 148) | Intervention v. Control Relative Risk Ratio (95% Confidence Interval) *p*-value |
|---|---|---|---|---|
| Having evaluation skill (*n* (%)) | Baseline | 44 (32.8) | 47 (31.8) | |
| | Follow-up | 70 (52.2) | 46 (31.1) | |
| Change in evaluation skill (*n* (%)) | Better | 37 (27.6) | 19 (12.8) | 2.47 (1.33, 4.59) *p* = 0.004 |
| | No change | 86 (64.2) | 109 (73.6) | (Reference outcome) |
| | Worse | 11 (8.2) | 20 (13.5) | 0.70 (0.32, 1.53) *p* = 0.370 |

### Primary outcome (eHEALS)

Table 2 shows the results regarding the eHEALS (means and standard deviations) and change between baseline and follow-up by group as well as differences between groups. There was a statistically significant difference between the intervention and control groups (Δ score change = 1.57; 95% CI [0.09–3.05]; *p* = 0.037).

### Secondary outcomes

Table 2 above shows the results regarding the HEL, which was not significantly different in the change between baseline and follow-up for either group (HEL: Δ score change = −0.08; 95% CI [−0.22–0.07]; *p* = 0.300). The proportional distribution of evaluation skill and its change after the intervention are displayed in Table 3. The intervention group was significantly likely to change from "no change" to "better" (RRR = 2.47; 95% CI [1.33–4.59]; *p* = 0.004).

### Results of the ancillary analysis

Nineteen participants dropped out of the study before the follow-up survey. Fourteen dropped out of the intervention group and five dropped out of the control group. Their missing scores on the outcome change variables were estimated using multiple imputation. Table 4 shows the estimation results of the regression analysis performed on

**Table 4 Intervention effect (Δ change[a] and Relative Risk Ratio) compared to control group using multiple imputation to create complemented dataset.**

| Results | Intervention v. Control Δ change[a] (95% Confidence Interval) | p-value |
|---|---|---|
| Score change on eHealth Literacy Scale (eHEALS) | 1.52 (0.05, 2.99) | 0.043 |
| Score change on Healthy Eating Literacy scale (HEL) | −0.06 (−0.21, 0.08) | 0.395 |

| Evaluation skill | Intervention v. Control Relative Risk Ratio (95% Confidence Interval) | p-value |
|---|---|---|
| Better | 2.27 (1.22, 4.24) | 0.010 |
| No change | (Reference outcome) | |
| Worse | 0.72 (0.33, 1.58) | 0.414 |

**Notes.**
[a] Score change of intervention group minus score change of control group.

the complemented data set. This result was almost the same as the result using the original data set.

The results of per-protocol analysis are shown in the (Tables S1 and S2). The estimate of the learning effect was larger than the result of the intention to treat analysis, but it followed the same trend as the intention to treat analysis.

## DISCUSSION

The results of this study indicate that eHealth literacy improved after a two-week e-learning program. This improvement was found in the eHEALS scores and in the participants' skill in selecting appropriate websites from search results. However, there was no significant change in health literacy regarding the HEL.

These results support previous studies' findings. For example, *Robinson & Graham (2010)* found that, after a 50-minute educational treatment, the eHEALS' scores of 18 participants increased from 19 to 32. Another previous study found that eHEALS' scores significantly increased in an elderly sample (assessed using Cohen's d) after an educational intervention (*Xie, 2011a*). In addition, the eHEALS scores in this sample significantly increased after intervention regardless of the educational or presentational method (*Xie, 2011b*). In the current study, the score improvement on eHEALS was not as large as in these previous studies, but the eHEALS scores increased by 2.31 points (standard deviation 7.27) after the intervention (Table 2).

Although the increased scores after educational intervention were consistent with previous studies, this study's effect sizes were relatively small. One reason for that inconsistency is that the learning effect on the e-learning platform might be weaker than the learning effect derived from other delivery methods. This possibility should be addressed by future research. Another reason for the difference might be that the participants did not learn sufficient content. In fact, 27 participants in the intervention group did not learn at all, and 10 participants learned only part of the content.

Furthermore, e-learning could be continuously employed after its content is prepared. Therefore, when it has a sufficient learning effect, it is a cost-effective way to enhance health. On the other hand, if the e-learning content were incorrect, it might be harmful, and, therefore, validation of content is important to the development of e-learning systems.

eHealth literacy is also influenced by differences in individual characteristics, such as age, educational attainment, healthcare experiences, Internet expertise, and so on (*Mitsutake et al., 2012*; *Mitsutake et al., 2016*; *Park, Moon & Baeg, 2014*), and eHealth literacy might be influenced more by face-to-face education than by e-learning (*Cox, Bowmer & Ring, 2011*; *Robinson & Graham, 2010*; *Xie, 2011a*; *Xie, 2011b*). Thus, it is necessary to determine the types of learning environments and methods (or combinations thereof) that might enhance eHealth literacy across diverse backgrounds.

### Strengths

This study has five important strengths. First, the randomized controlled trial demonstrated that e-learning is an effective way to educationally intervene because any causal inference would not be influenced by confounding bias. Second, the proportion of responses in the follow-up survey was very high (93.7%), which minimized the effect of selection bias. Third, in the complemented dataset, eHEALS scores and evaluation skill increased due to the intervention. This indicates there was little influence of dropouts. Fourth, not only the subjective score (eHEALS), but also the objective score (evaluation skill), improved due to e-learning. Fifth, the eHEALS rose significantly, but the HEL scale did not. This suggests that the increased scores were scarcely influenced by the Hawthorne effect.

### Limitations

Regarding the measures used in the analysis, the variables other than eHEALS and HEL were not validated, and the participants' evaluation skills might not have been correctly evaluated. However, the interpretation of the results was not distorted because statistically significant results were found on the primary outcome (eHEALS), which was validated. Using self-report data to assess outcomes might cause non-differential misclassification, but when this type of misclassification occurs, it does not influence the point estimates or widen the confidence intervals. Therefore, using self-report data in this study did not influence the interpretation of its results. Last, because the learning effect was evaluated after a short two-week period, it could not be determined whether the effect of e-learning was retained for a longer time. Follow-up studies that cover longer periods are recommended.

### Generalizability

The results of this study have limited generalizability because it targeted participants with an existing interest in health literacy and e-learning. The tokens distributed to the participants might have encouraged the intervention group to be more motivated, and the e-learning participation rate was considered to be high. If the tokens had not been distributed, the participation rate would have been considered low in e-learning education for the general population. Therefore, e-learning in the general population might yield a result different from that of this study.

## CONCLUSIONS

Although this study has some weaknesses, its results using RCT could demonstrate that e-learning education had a positive effect on eHealth literacy for Japanese Internet users. Furthermore, this study suggests that e-learning is an effective way to improve eHealth literacy.

## ACKNOWLEDGEMENTS

I am grateful to all of the study's participants for their participation and to Macromill, Inc. for implementing the online survey. I would like to thank Editage (http://www.editage.jp) for English language editing.

### Funding

This study was supported by Milk Education Research Council (Subcommittee of Japan Milk Academic Alliance). The funders had no role in study design, data collection and analysis, decision to publish, or preparation of the manuscript.

### Grant Disclosures

The following grant information was disclosed by the author:
Milk Education Research Council.

### Competing Interests

The author declares that there are no competing interests.

### Author Contributions

- Toshiharu Mitsuhashi conceived and designed the experiments, performed the experiments, analyzed the data, contributed reagents/materials/analysis tools, prepared figures and/or tables, authored or reviewed drafts of the paper, approved the final draft, submitted documents to the Ethics Committee.

### Human Ethics

The following information was supplied relating to ethical approvals (i.e., approving body and any reference numbers):

This study was approved by the Okayama University Graduate School of Medicine, Dentistry and Pharmaceutical Sciences and Okayama University Hospital, Ethics Committee (approval number K1707-025).

### Data Availability

The raw data are provided in the Supplemental Files.

### Supplemental Information

Supplemental information for this article can be found online at http://dx.doi.org/10.7717/peerj.5251#supplemental-information.

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
