# Peer review of "Effects of two-week e-learning on eHealth literacy: a randomized controlled trial of Japanese Internet users"

_PeerJ, doi:10.7717/peerj.5251_

## Round 0.1 · original submission · Major Revisions

Please respond to each point raised by the two reviewers, to explain how you have addressed their points or why you disagree. In addition, I have a couple of suggestions.

- I would recommend including effect sizes, such as Cohen's d, to illustrate the results.
- I would clarify that the primary analyses are the observed scores, not the imputed dataset
- I agree with Reviewer 1 that the attitudes measure needs further description and justification. Is there are reliability or validity data to support it? Eg. test-retest reliability in the control group, correlation with the eHEALS score?
- I couldn't follow who performed the randomisation/assignment to groups. Was this automated?

·

Basic reporting

The basic reporting of this manuscript is good and sophisticated.

Experimental design

The author reported that the short-period e-learning has a positive effect on the health literacy in a randomized control study. The author evaluated the internet users’ health literacy with the eHealth Literacy Scale, which has been validated in the prior studies. The score change of the eHealth Literacy Scale in the training group was significantly higher than that of the non-intervention group. The author evaluated the other scales. The author concluded that the e-learning is useful to learn the health literacy on the web.
However, Several concerns in method and terminology remained as below.
1. The reviewer had severely concerned that the score of Attitude. Firstly, the reviewer is in doubt as to whether this question about osteoporosis is appropriate. Secondary, the scoring system of attitude is not appropriate because the sharing the wrong information widely did not equal to investigate the scientific evidence. Third, the score of attitude has never been validated in a prior study. The score of attitude had several critical problems, so the reviewer recommended the attitude should be expunged in this manuscript.
2. The author defined the subjects as having the evaluation skills person, who had selected the government’s website. Van Deursen and Van Dijk elaborated four types of internet skills in their prior study. These are termed operational, formal, information, and strategic internet skills (JMIR 2011). They mentioned as internet skills not only to evaluate information source but also to make the right decision to reach their goals. Why did the author ignore the other skills such as operational, formal and strategic skills?
3. The HON criteria (Boyer C. Comput Bio Med, 1998), The DISCERN score (Charnok D. J Epidemiol Commun H, 1999) and JAMAcriteria (Silberg WM, JAMA. 1997) are known as the tools to evaluate the quality of health information of the internet. The reliability of the individual website did not depend on only the information provider but also authoritative, attribution, transparency and financial disclosure. However, the author evaluated the reliability of contents with only the address of the website. Could you show the evidence or reference that the subjects who selected the go.jp or co.jp website had the internet skills?
4. The e-learning contents played the important roles in this study. Could you show the examples of the e-learning teaching materials? If possible, the author should demonstrate these contents translated into English as Supplementary data. If impossible, the author should describe the content more in detail.
5. The author mentioned, “The HEL was employed to assess change to health literacy other than a change to eHealth literacy.” Could you explain the difference between the HEL and eHealth Literacy Scale? The author should demonstrate these scales as figures or tables because these scales were unfamiliar with the audience.
6. If the e-learning contents did not have information about healthy eating literacy, it is natural that the value of HEL did not change after training. The author should show their e-learning contents, which also included the health eating literacy or demonstrate the reference, which the score of HEL can change after receiving some kind of e-learning.
7. The total 37 subjects (27.6%) were not able to complete the e-learning program in the intervention group. This author should mention the fact in the result section because the rate of adherence would affect the results. It may prefer to analysis separately in each case.
8. Line 185-186
The meaning of the sentence is unclear. Please rewrite it.
9. Line 186-187
When the subject selected the governmental website, the subject had the evaluation skill. In results, the author divided the three groups, “better”, ”no change” and “worse” in table 3. There is no definition of the groups in this manuscript. The author should define these groups.
10, 11. Line 203-208
10. Was the primary outcome the change of the eHealth Literacy scale? Similarly, were the second outcomes the change of the HEL, the attitude and the skills?
11. What kind of the statistical test did the author use when the author compared the change in scores? (Student t-test? Mann-Whitney U test?) . Please describe the details of statistical analysis.

Validity of the findings

This study could demonstrate that e-learning education had a positive effect on eHealth literacy for Japanese internet users. The conclusion is redundant. It need to rewrite it.

Additional comments

Abstract
Line 27-32 and 35
 “There were no large differences at baseline between the intervention and control groups in the eHealth literacy scale or skill level for evaluating search results.”
1. Is “in the eHealth literacy” in this sentence “the Healthy Eating Literacy Scale” right?
2. What did the author mean skill “level”? Please define the skill “level” in this manuscript.

Line 32-33
There is no data to improve both outcomes at the follow-up.

Line 34-37
In this manuscript, the author demonstrated the significant statistical difference of the change for eHealth Literacy scales. However, it is unclear whether the statistical difference has clinically meaningful. In addition, the author could not show the difference of the secondary outcomes. Please rewrite this section.


Results
Line 238-240 and Line 247-249
The reviewer did not understand the results of the relative risk ratio (RRR). Because the meaning of “no change” had two patterns as follows, 1) the subject who had had no skill from the beginning to the end, 2) the subjects who had had enough skill from the beginning to the end. How did the author treat the “no change” group in their analysis?

If the author would like to demonstrate the relative risk, the author should use the data at follow-up point.
Getting the skill No skill sum
Intervention 70 64 134
Control 46 102 148

Relative risk: (70/134) / (46 /148)=1.68 [95% confident interval 1.26-2.24],
Fisher exact test: P=0.0004

Discussion
Limitation and Generalizability were written well.

Conclusions
Line 300-302
This sentence,“Although the e-learning~” is tediousness. This content of a sentence is merely a repetition of what was already mentioned in Introduction.

Line 304-314
It seems that these sentences are not a conclusion. The author should move these sentences to the Discussion section.

Line 314-315
This sentence is very assertive. This study provided with the interesting results. However, There are several limitations at the same time.
“This study could demonstrate that e-learning education had a positive effect on eHealth literacy for Japanese internet users.”

Figure and Tables
Table 1
1. It is preferable that P value between intervention group and control group is described.
2. The reviewer cannot understand the meaning of “Self-rated health”. Please explain the term in this manuscript.

Table 2 and Table 4
“a score change of intervention~” made no sense. Where is the “a” in this table?

Minor comment
Line 132 “In a previous study”
Please describe the reference.

·

Basic reporting

Thank you for the opportunity to review this easy to read manuscript it would benefit from additional revision to strengthen its contents. While this article adds to the literature regarding ehealth literacy acquisition by Internet users, some of the literature in the introduction is older than 5 years and use of more contemporary articles could strengthen this section of the manuscript, especially as they are not seminal work. Update of references would improve relevancy to readers. Removal of the colloquial language and explanation of acronyms would also improve readability. Please see general findings for specific examples. The conclusion needs to be re-written to ensure it does not contain any new information that requires referencing, the importance of this information needs to be included earlier in the manuscript.

Experimental design

The experimental design, including recruitment pool, limitations and issues with generalisability are mentioned and justified.

Validity of the findings

This study is clearly described and could be replicated by others.

The conclusion needs to be re-written to ensure it does not contain any new information that requires referencing, the importance of this information needs to be included earlier in the manuscript.

Additional comments

To improve readability and to clarify the definition of ehealth literacy line 13, could be improved by adding the word ‘health’ after ‘accuracy’. Line 24, were the score computed or analysed? Use of acronyms such as WELQ (line 50) US (line 59), eHEALS (line 131) needs to be described in full the first time used. Removal of colloquial language, lines 51, 77, 131, 267 and 309 would improve readability. A range of tenses is used. I am unsure first person improves readability of the manuscript. Some of the references are quite old and are not seminal and more contemporary literature may strengthen the article. For example lines 41, 43, 49, 59-61, 71.

The innovation of using an RCT that identified and uses a modified form of CONSORT is a useful addition to the literature.

---

## Round 0.2 · Minor Revisions

Thank you, you have adequately addressed my suggested changes. Both reviewers agree that the revised paper is much strengthened but a few minor changes are required before the paper can be accepted. Please see comments from Reviewer 1 and 2 below. I do not feel that reviewer 2's comment under 'Basic reporting' is necessary to address.

·

Basic reporting

Thank you for revising the manuscript “Effects of two-week e-learning on eHealth literacy: a randomized controlled trial of Japanese Internet
users ”
The author almost appropriately responded the issues of the manuscript. However, The reviewer1 have one question for statistical process.

Experimental design

Line 215-219

The author used the linear regression analysis and multinomial logistic regression analysis in this study.
In this study design the repeated measured ANOVA is prefer to the linear regression analysis. Why did the author use the linear regression analysis?

Validity of the findings

no comment

Additional comments

The manuscript improved especially in "material and method".

·

Basic reporting

I have used the track change version to review this resubmission:

The revisions have improved readability and understanding of the research and findings. Clarity has improved and the authors have addressed the issues raised. Sharing of the elearning content in English is valuable. On revision, I also concur with removal of the attitude information as it strengthens the remaining information.
Line 71, about levels of health literacy needs referencing, currently it is a broad statement with no evidence. Alternatively the full stop could be removed and "as evidenced by a German survey" could be inserted instead.

Experimental design

Inclusion of the content in English is valuable. Removal of the attitude information strengthens the article. Line 46 that ends with Hawthorne effect, needs another sentence to conclude the abstract. I believe ending with the Hawthorne effect idea weakens the tenet of the abstract. Line 122 needs a comma after 'aware of, and'; line 138 is 'participant data', 'not participants'.

Validity of the findings

Clarity has been improved by deletion of the attitude questions and provision of supplementary materials. This section has minor edits including removal of brackets around the participant withdrawal or non-completion numbers on line 306-307. The addition as suggested of the Cohen's d is noted.

The authors still need to provide a strong conclusion that represents their findings. They have understated the significance of the research as it is rare to have an RCT of this nature. While the research does have limitations they are stated above in the generalizability section.

Additional comments

The revisions have strengthened the article. Inclusion of information regarding the Hawthorne effect and rationales for use of HEL strengthens the article and improves readability. The inclusion of the elearning content is valuable to readers. Removal of the attitude information simplifies the study and strengthens the findings. Ending the abstract with a summative sentence and ensuring the conclusion reflects the findings and discussion would also strengthen the article.

---

## Round 0.3 · accepted · Accept

Thank you for the revised manuscript and response to reviewers' comments. The manuscript is now acceptable for publication.